# Effect of Bioactive Packaging Materials Based on Sodium Alginate and Protein Hydrolysates on the Quality and Safety of Refrigerated Chicken Meat

**DOI:** 10.3390/polym16233430

**Published:** 2024-12-06

**Authors:** Svetlana Merenkova, Oksana Zinina

**Affiliations:** Department of Food and Biotechnology, South Ural State University (National Research University), 76 Lenin Avenue, Chelyabinsk 454080, Russia; zininaov@susu.ru

**Keywords:** alginate-based films, bioactive packaging materials, protein hydrolysates, chicken meat, oxidative processes, fatty peroxide value, fatty acid value, total microbial count, color characteristics

## Abstract

The purpose of this study was to evaluate the potential of alginate-based packaging materials with the incorporation of protein hydrolysates to improve the safety and quality of chicken meat during storage. Physicochemical parameters, microbiological indicators, and color characteristics of chicken meat packaged in bioactive films were determined. We observed a significant increase in moisture content for samples in polyethylene films (by 10.5%) and decrease for the samples in alginate-based films by 5.3%. The highest mass losses were found for the sample without packaging material (20.4%) and for the samples wrapped in alginate films (15.9–17.9%). When packing meat samples by immersion method, a gradual decrease in weight was found (up to 9.1%). On the 7th day of storage, the pH value of the control sample reached 6.55, while for the samples in bioactive alginate-based materials pH level was 6.0–6.15. The most pronounced oxidative processes were observed in the control meat sample (5.1 mmol (12O_2_)/kg). The application of bioactive alginate-based films led to a significant reduction in fatty peroxide value by 56.2%. The total microbial count in the meat samples packaged in bioactive films was 3.5–5 times lower than in the control sample. Chicken meat wrapped in alginate-based films with protein hydrolysates maintains more stable color characteristics, the lightness index (L) decreased to 37.5, and the redness index (b) increased to 3.4 on the 7th day of storage.

## 1. Introduction

Poultry meat production remains consistently the highest among all types of meat. In the recent decade, its production has shown significant growth due to lower cost, shorter production cycles and growing demand worldwide. Global poultry meat output is forecasted to reach 150 million tonnes in 2024, a 2.5 percent increase over 2023, with the largest production growth in all major producing countries, particularly in China, the European Union, the United States, Pakistan, India, Egypt, and Brazil [1].

According to the data from the Food and Agriculture Organization, the total poultry presence in the world amounted to about 27.9 billion heads in 2019. Chicken accounts for the largest share of this presence—about 93 [2]. Chicken has become one of the most popular meat products in the world due to its low-fat content, high protein content, high nutritional value, and easy digestion. Chicken meat provides significant amounts of essential vitamins and minerals that are crucial for maintaining various body functions, such as energy production, immune support, and bone health [3].

Due to the specifics of its chemical profile, poultry meat is a food system in which microbiological and enzymatic processes occur even at low temperatures (from 0 to +4). These processes cause alterations in the structure of proteins and lipids, reduce the nutritional value of meat, and lead to the accumulation of biochemical reaction residues and microbiological spoilage products that can be hazardous to human health [4]. During the production, storage, distribution and transportation processes, chicken meat could be easily spoiled due to the combined effects of atmospheric oxidation, microorganisms and endogenous enzymes. Enzymes, proteins, and lipids are the main factors affecting the quality of chicken meat [5].

High water content in poultry meat can result in rapid growth of numerous microorganisms, pathogenic bacteria, and in association with high activity of endogenous enzymes, can cause significant changes in the physicochemical and microbiological characteristics of meat, which markedly affects its quality. Even under low-temperature freezing and refrigeration conditions, lipoxygenase is still active and catalyzes fat oxidation and rancidity in the presence of oxygen, causing the modification of color and smell of meat and reducing its quality [5,6].

Nowadays, it is highly relevant to study the preservation of food raw materials, especially raw materials subject to fast microbiological spoilage, which includes poultry meat. There are several approaches to increase the safety of meat raw materials and prolong their shelf life; the most used ones include the use of low-temperature processing (cooling; freezing), the application of chemical preservatives, and application of new packaging techniques.

During the cooling and refrigeration process, the temperature of poultry meat must be controlled at level 0~8 °C, which can inhibit the activity of enzymes, the growth and reproduction of microorganisms, which is beneficial to the maturity of chicken meat and prevents ice crystals from damaging muscle tissue. Refrigerated fresh chicken meat has more advantages than frozen meat in terms of taste, flavor, freshness and nutritional value [7,8,9]. However, during cooling the poultry meat, it is essential to control refrigeration conditions to prevent spoilage, ensure product safety and preserve the flavor and nutritional quality [10]. The effect of cold temperature storage on meat quality remains a topic of ongoing scientific investigation aimed at enhancing the product’s shelf life [11].

Packaging is crucial for the transportation and storage of poultry meat, preventing the product from being exposed to unfavorable environmental factors—damage, deterioration, and microbial contamination—and helping to prolong and preserve the freshness of the product. According to research data, appropriate food packaging materials prevent the loss of moisture and nutritional compounds during product storage [12]. Currently, the market offers modified atmosphere and vacuum packaging, active packaging, and antibacterial packaging [13].

With the development of preservation technology, new packaging technologies are constantly emerging. Active packaging maintains food quality and extends shelf life by changing the packaging environment, while improving food safety and sensory properties. Active packaging mainly includes oxygen removal, moisture absorption and regulation, carbon dioxide control, and antimicrobial systems [13,14]. Some researchers also propose composite biodegradable materials, nanocomposite packaging and edible films made of natural substances [15,16].

The most advanced approaches to prevent undesirable processes in poultry meat during storage and transportation include the use of packaging with bioactive compounds. Moreover, there is a significant number of scientific findings which describe encouraging effects in the application of composite materials based on organic biopolymers with the introduction of bioactive ingredients with antimicrobial, bacteriostatic, and antioxidant properties [17,18,19,20].

Active food packaging is a new approach to sustainable waste management and the preservation of food quality and safety. Eco-friendly food packaging has recently received more attention as there is a growing concern for the development of new types of renewable and biodegradable materials. The incorporation of natural compounds with functional activity in their composition guarantees increased food safety. Active packaging aims to improve the quality and safety of meat products through the interaction of packaging, product and environment. The inclusion of active compounds such as antioxidants, antimicrobial agents, and essential oil is one of the most promising methods for the development of active food packaging [21,22,23,24,25].

Protein hydrolysates are described as a result of enzymatic or acid hydrolysis of native protein components. It is a complex organic system containing residues of polypeptide components, peptides, and free amino acids. In our previous investigations, the technology of obtaining protein hydrolysates from poultry processing by-product was described [26], their biochemical composition was studied [27], and the antioxidant and bacteriostatic properties were proved [28]. In addition, a series of experiments were performed to investigate the technology of introducing protein hydrolysates into the composition of bioactive packaging materials based on plant polysaccharides (pectin, agar, sodium alginate). We established an improvement in the mechanical, structural, and physico-chemical properties of films based on sodium alginate with the incorporation of protein hydrolysates, which proved high compatibility of the hydrolysate with the alginate matrix structure [29].

Numerous publications have demonstrated the potential of protein hydrolysates in the composition of biodegradable packaging materials, which positively affect the safety of food, prevent the development of oxidative and hydrolytic reactions during the storage of food products, as well as inhibit the growth of microorganisms and bacterial pathogens [30,31,32].

However, more research is needed to prove the potential of protein hydrolysates generated by enzymatic treatment of poultry by-products for application in bioactive biodegradable packaging materials. Thus, the purpose of this study was to evaluate the potential of sodium alginate-based packaging materials with the incorporation of protein hydrolysates to improve the safety and quality of poultry meat during storage. Since the activity of protein hydrolysates depends on the level of contact with the surface of the product, we applied two approaches for packaging poultry meat in the developed biodegradable materials wrapping in dried film and immersion in the composite solution.

## 2. Materials and Methods

### 2.1. Raw Materials and Ingredients

The objects of the study were samples of chicken meat (filetfilet) (produced by Chebarkul Poultry LLC, Chelyabinsk, Russia). The meat was purchased on the first day after slaughter and cooling to a core temperature of 4 ± 2 °C.

Sodium alginate (Ingredico LLC, Moscow, Russia), glycerin (Iodine Technologies and Marketing LLC, Moscow, Russia), and protein hydrolysate as active components were used to obtain the film. Protein hydrolysate was obtained by microbial fermentation of broiler chicken gizzards with propionic acid bacteria. The technology is described in [26,27].

Protein hydrolysate was obtained by microbial fermentation of broiler chicken gizzards with propionic acid bacteria.

### 2.2. Sample Preparation

The research was carried out in the laboratories of the Department of Food and Biotechnology of South Ural State University (National Research University) (Chelyabinsk, Russia).

The films were obtained as follows: 1.5% sodium alginate was added to water heated to 40 °C with constant stirring on a magnetic stirrer. After 20 min of stirring, a 3% glycerin solution was added and a 1% protein hydrolysate solution, then poured onto metal sheets and dried at 30 °C for 24–28 h. In addition, films were made similar to the technology described above, but without the addition of protein hydrolysate (Figure 1).

### 2.3. Packaging of Fresh Chicken Meat

Fresh chicken meat was divided into portions of 60 g and was wrapped in the previously made films (5 × 18 cm^2^). Two methods of packaging chicken meat were used: wrapped in a film and immersed in a liquid film composition. Unwrapped meat acted as the control, as well as chicken filetfilets packed in PET film, and the experiment was carried out in triplicate. The chicken meat samples (for codes, see Table 1) were stocked inside plastic boxes with a screw cap and placed under refrigeration (4 ± 2) °C for 7 days. Each batch was picked and characterized at 0, 3, and 7 days of storage. Physicochemical and microbiological parameters, as well as indicators characterizing the oxidation of the lipid fraction, were studied at 0, 3, and 7 days of storage.

### 2.4. Experimental Methods

#### 2.4.1. Moisture Content (MC)

For moisture determination, based on the AOAC method [33], 5 g of poultry meat was weighed in a previously weighed crucible. Samples were placed in an oven at 103 ± 2 °C until constant weight. For moisture content calculation, Equation (1) was applied, where P_1_ represents the weight of the crucible + meat before oven, P_2_ the weight of crucible + meat after oven and P_f_ the weight of the crucible alone.
(1)$$\mathrm{MC}=\frac{\left(P_{1}-P_{2}\right)}{\left(P_{1}-P_{f}\right)}\times 100.$$

The mass loss was determined gravimetrically as the difference in the mass of the chicken meat sample before and after storage and expressed in %.

#### 2.4.2. pH Level Analysis

The pH analysis is based on the determination of hydrogen ion activity. The methodology was performed according to the AOAC [33], where 5 g of poultry meat was weighed and 50 mL of deionized water at 40 °C was added. The mixture was agitated for 15 min. After filtration using qualitative filter paper, pH was determined in the solution with a pH digital meter (CRISON micropH 2001, Barcelona, Spain), previously calibrated with pH 4 and pH 7 buffer solutions.

#### 2.4.3. Investigation of Microbiological Indicators

Three microbiological indicators were determined using microbiological rapid tests “Petritest” (NGO “Alternative”, Saratov, Russia): mesophilic aerobic and facultative anaerobic microorganisms (MAFAnM), total aerobic psychotropic microorganisms, and *Enterobacteriaceae*. Appropriate dilutions were made for each meat sample and applied to Petri dishes (1 mL) with a suitable medium. Plate Count Agar (PCA) was used to test total aerobic mesophilic microorganisms and total aerobic psychotropic microorganisms. Dry chromogenic medium was used for detection and identification of *Enterobacteriaceae*. Sowing for Petritests, thermostating and processing of the results were performed in accordance with the manufacturer’s recommendations set out in the Guidelines 4.2-022-20165 [34]. Mesophilic aerobic and facultative anaerobic microorganisms (MAFAnM), and *Enterobacteriaceae* were counted on Petri dishes after incubating for 24 h at 26 ± 1 °C, while total aerobic psychotropic microorganisms were counted after incubating for 168 h at 7 °C. The results are expressed as log CFU (colony forming units)/g meat.

#### 2.4.4. Fatty Acid Value

The method is the titration of free fatty acids with a solution of potassium hydroxide. A petroleum ether-ethanol blend (1:2) was stirred and neutralized with phenolphthalein and 0.1 M potassium hydroxide until a persistent deep red appeared. Fat was separated from meat samples and was dissolved in the ether-ethanol blend within a flask, titrated with potassium hydroxide until the red color persisted for over a minute, calculating the fatty acid value as the mean of duplicate repeatability tests. The fatty acid value (*FA*) is expressed in milligrams of potassium hydroxide consumed to neutralize the free fatty acids contained in 1 g of fat, mg(KOH)/g, calculated by Equation (2):(2)$$FA=\frac{V\times 5.61}{m}.$$

In the formula, *V* is the volume of potassium hydroxide solution with molar concentration of 0.1 mol/dm^3^ consumed by titration, (cm^3^); 5.61 is the mass of potassium hydroxide contained in 1 cm^3^ of potassium hydroxide solution with molar concentration of 0.1 mol/dm^3^, (mg); *t* is the mass (g) of the fat sample analyzed.

#### 2.4.5. Fatty Peroxide Value

Preparation of reagents

Sodium thiosulfate solutions (0.01 and 0.002 M) were freshly diluted, and 1:1 acetic acid-chloroform mixture was prepared. 1% starch solution was prepared, and saturated potassium iodide was created in darkness, ensuring iodine absence.

Fat samples were dissolved, reacted with iodide, then titrated with thiosulfate, alongside a control devoid of fat. The peroxide value was calculated, with the mean of duplicates as the result. Fatty peroxide value (FP), mmol (12O_2_)/kg, calculated using the following Equation (3):(3)$$FP=\frac{1000\times K\times \left(V-V_{1}\right)\times c}{m}.$$

In the formula, *K* is the correction factor for the titer of the sodium thiosulfate solution; *V* is the volume of the sodium thiosulfate solution with a molar concentration of 0.01 mol/dm^3^, used for titration of the released iodine, (cm^3^); *V*_1_ is the volume of sodium thiosulfate solution with a molar concentration of 0.01 mol/dm^3^ consumed by titration in the control experiment, (cm^3^); *c* is the molar concentration of sodium thiosulfate solution, (mol/dm^3^); *m* is the fat mass, (g).

#### 2.4.6. Fatty Iodine Value

The essence of the method is the following. The analyzed fat sample is dissolved in a solvent and the Wijs reagent is added. After a certain time, potassium iodide and water are added, and the released iodine is titrated with a solution of sodium thiosulfate (0.1 mol/dm^3^).

The sample is weighed into a flask and a solvent is added (a solution of cyclohexane and glacial acetic acid (1:1)). 25.0 cm^3^ of Viisa reagent is added with a pipette. The flask is closed with a stopper, the contents are mixed by rotating the flask and the flask is placed in a dark place, kept for 1 h.

At the end of the reaction, 20 cm^3^ of potassium iodide (mass concentration 100 g/dm^3^) and 150 cm^3^ of water are added. The contents of the flask are titrated with a solution of sodium thiosulfate (0.1 mol/dm^3^) until the iodine-induced yellow color practically disappears. A few drops of 1% starch solution are added and titration continues with vigorous shaking until the blue color disappears. The Fatty Iodine Value (*FI*), g/100 g is calculated by Equation (4):(4)$$FI=\frac{12.69\times \left(V-V_{1}\right)\times c}{m}.$$

In the formula, *c* is the concentration of sodium thiosulfate solution, mol/dm^3^; *V* is the volume of sodium thiosulfate solution consumed in the control titration, cm^3^; *V*_1_ is the volume of sodium thiosulfate solution consumed during titration of the sample, cm^3^; *m* is the sample weight, g.

#### 2.4.7. Investigation of Color Characteristics

The color characteristics were studied using the NR60CP colorimeter (Shenzhen Sheen Technology Co., LTD, Shenzhen, China). Before using the NR60CP colorimeter, the device was calibrated using a white standard plate (*L** = 96.77, *a** = 0.11, *b** = −0.71), which was also used as a background for measuring the color characteristics of films (lightness (*L**), redness (*a**) and yellowness (bluishness) (*b**). The Total Color difference (*ΔE*) and Color Intensity (*Chroma*) were calculated in Equations (5) and (6):(5)$$\Delta E=\sqrt{{\left(L^{*}-L\right)}^{2}+{\left(a^{*}-a\right)}^{2}+{\left(b^{*}-b\right)}^{2}}\;,\;$$
(6)$$Chroma=\sqrt{a^{2}+b^{2}}\;$$

In the formula, *L**, *a**, *b** are the standard values of the white plate color parameters; *L, a, b* are the values of the film color parameters.

#### 2.4.8. Investigation of Rheological Characteristics

Rheological characteristics, including general, elastic, and plastic types of deformations of chicken meat, were determined using the Structurometer ST-2 (St. Petersburg, Russia), based on mathematical processing of the exponential relaxation curve of mechanical tension arising on the cylindrical indenter during its penetration into the meat. The principle of operation of the device is based on the measurement of mechanical stress on the indenter nozzle when it is introduced at a specified speed into the prepared sample. The following mode of operation of the texture analyzer was set: moving the indenter down to the contact with the sample at a speed (*V*σ = 0.5 mm/s) with a force (*F**k* = 7 g); deforming the meat sample with the indenter to a force (*F**m**a**x* = 500 g).

#### 2.4.9. Statistical Analyses

The analyses were performed in three replicates. The results were expressed as the mean values of the three replicates ± the standard deviation. Probability values of *p* ≤ 0.05 were taken to indicate statistical significance. The data were analyzed via one-way ANOVA analysis of variance using the free web-based software presented by Assaad et al. [35].

## 3. Results and Discussion

### 3.1. Moisture Content

The moisture content in food can affect both the sensorial quality and the stability of the product, since the deterioration process can correlate with the content of water. Moisture content in chicken filet can vary from 64 to 68%, and higher moisture level triggers active multiplication of microorganisms and enzymatic processes during refrigerated storage [36].

In this experiment, the initial moisture content was 68%, and after 7 days of storage, the moisture content in the control sample gradually decreased by 3.8%. The moisture content values were different when packing chicken meat samples in various types of films. Thus, there was a significant increase in humidity in polyethylene-based films (by 10.5%), which is due to the low air and vapor permeability of this type of packaging material and, first of all, creates the risk of adverse processes in meat. When packing meat samples by wrapping in alginate films, a decrease in moisture was observed (by 3.1–5.3%) due to the excellent hydrophilic characteristics of alginate films and their significant vapor permeability, which was demonstrated in our previous studies [37]. Promising results were obtained using alginate films applied by immersion of poultry meat in a composite solution. For this approach, insignificant fluctuations in moisture content of chicken meat samples were observed within the range of 0.5–1% (Figure 2).

It is reasonable that mass losses for chicken meat correlated with the dynamics of moisture content of the samples. The highest mass losses were found for the control sample (without packaging material)—up to 20.4% and for the samples wrapped in alginate films (15.9–17.9%). When packing meat samples by immersion method, a stable gradual decrease in weight by 8.5–9.1% was observed. This trend is due to a denser contact of the packaging film, and less absorption of moisture from muscle tissue, while maintaining sensory properties and inhibition of microbiological and oxidizing processes (Figure 2).

Silva et al. (2023) have proved that the unwrapped poultry meat demonstrated an increase of up to 3% in moisture content during storage. On the other hand, all the meat samples wrapped in bioactive films presented a decrease in the moisture because polysaccharide chitosan has hydrophilic properties and absorbed some of the meat water [20].

### 3.2. pH Level

The pH value regulates the reactions that take place in food systems during the storage period; therefore, it is important to monitor this value to maintain the safety and good preservation of the product. The pH of the muscles after slaughtering the animal decreases from about 7.2 to about 5.5, as glycogen is converted into lactic acid. The pH values of fresh chicken meat usually range from 5.7 to 6.0, depending on the quality of the meat and the conditions of processing and storage [38].

The initial pH value for the meat sample was 5.85–5.9, reaching a pH value of 6.55 for the control sample on the 7th day of storage (Figure 3). An increase in pH values may be associated with the growth of microorganisms producing low molecular weight components [35]. Other results were obtained using alginate packaging of meat, the indicators of which changed slightly compared to the initial values, while the samples for which alginate-based packaging materials with the addition of protein hydrolysate were used showed the most stable results (pH on 7th day did not exceed 6.0–6.15).

The results are consistent with the studies of other scholars who observed stable pH values of poultry meat when stored in packages containing bioactive components [10,39].

### 3.3. Lipid Oxidation Indicators

Poultry muscle tissue contains monounsaturated and polyunsaturated fatty acids which are highly oxidized by oxygen and are also affected by lipolytic enzymes. Therefore, meat which is stored without packaging is subject to rapid oxidative spoilage, which is confirmed by the results of the experiment conducted. The generation of peroxidation products simultaneously leads to a decrease in Fatty Iodine Value. Further oxidation of peroxides leads to the formation of carbonyl acid and other toxic compounds.

To monitor the lipid oxidation of the samples, the Fatty Acid Value, Fatty Peroxide Value and Fatty Iodine Value (FA; FP; FI) were determined. The most pronounced oxidative processes were established for the control sample of meat without the use of packaging materials, which demonstrated the highest FP values, while the application of alginate films with bioactive protein hydrolysate contributed to the inhibition of oxidative and hydrolytic processes. Thus, the FP values for the samples WF-AF + PH and IF-AF + PH decreased by 52.9–56.2% after 7 days of storage compared with the control, and the FI values remained at the level of 17–18 g/100 g, which indicates the preservation of unsaturated acids for these samples by more than two times compared with the sample without packaging (Table 2).

During the storage of poultry meat, endogenous lipases cause hydrolysis of triglycerides leading to the accumulation of fatty acids in muscle tissue, which is reflected in an increase in the Fatty Acid (FA) value. For meat samples without packaging, the highest FA was established after 7 days of storage. The use of all types of packaging materials resulted in the inhibition of enzymatic processes and a decrease in FA, but the lowest FA values were detected for samples packaged in alginate-based films with protein hydrolysate (32.9% lower compared to an unwrapped sample) (Figure 4). The obtained results confirm the antioxidant characteristics of packaging materials when protein hydrolysate is incorporated into their composition.

### 3.4. Microbiological Indicators

Since chicken meat is the most frequently purchased food product, chilling at low temperatures (0… +4) °C is the most commonly used way for its storage and transportation. Meat is a suitable nutrient medium for bacteria, as it contains a large amount of free moisture, amino acids, peptides and carbohydrates. The presence of bacteria on the surface of meat depends on environmental conditions and initial contamination. It is believed that poultry meat can be stored for up to 6 days in chilled conditions. Spoilage of refrigerated meat is associated with psychotropic microorganism’s growth as well as biochemical and enzymatic deterioration [10,40].

The results presented in Table 3 show that films with the addition of protein hydrolysate had an inhibitory effect on bacterial growth, which is associated with the presence of active peptides in its composition that have a destructive effect on the cell membrane of bacteria causing their death [29]. Thus, on the 7th day of storage, the MAFAnM for meat samples packed in films with bioactive hydrolysate was 3.5–5 times less than in the control sample. The highest microbial contamination was found for the sample packed in a polyethylene film, which could be associated with poor air permeability of the packaging material and accumulation of excess moisture which caused rapid growth of bacteria. The microbiological limit for MAFAnM in chilled meat, as established by the Technical Regulations of the Eurasian Economic Union (EAEU TR 051/2021), is 1.0 × 10^5^ CFU/g [41]. The MAFAnM values for the control sample and the sample in a polyethylene film exceeded the limit values. In general, all the alginate-based films were effective in controlling the microbial growth at chicken meat surface showing the capacity to increase shelf life of poultry meat.

During the study, it was demonstrated that bioactive films contribute to the prevention of Aerobic psychotropic microorganisms and *Enterobacteriaceae* accumulation. The values of these indicators were significantly reduced compared to control samples and samples in PET film after 7 days of storage. Comi et al. (2017) have proved that *Enterobacteriaceae* family were the most reported microorganisms occurring in raw meat. The authors noted that during the 6-day monitoring, most of the detected microorganisms were able to reproduce. Therefore, it is likely that, though at a reduced speed, many of the bacteria were able to grow during refrigeration [42].

### 3.5. Color Characteristics

Sensory parameters are the most important factors when choosing a product by the consumer, and color can serve as an indicator of oxidative changes during spoilage of raw materials. The results of a quantitative assessment of the color characteristics of chicken filet show that during storage, the surface of meat samples becomes darker (Table 4). The redness index (a) increases in all samples simultaneously with a decrease in the lightness index (L). However, changes are more pronounced in uncoated chicken filet samples, which is explained by the active interaction of the meat surface with air oxygen and intensive oxidative processes of meat pigments. The data obtained on changes in the color characteristics of chicken meat during storage are consistent with the results of other scientists. Thus, Orkusz et al. (2024) noted that during refrigerated storage of turkey filets, the parameters of lightness (L) and redness (a*) gradually decreased over 6 days of storage compared with the initial values [10].

In the meat samples under study, the lightness index (L) in the uncoated sample decreased by 34.5% (*p* ≤ 0.05) on the 7th day of storage compared to the initial value, while in the coated samples it decreased by 22.07–27.8% (*p* ≤ 0.05). The redness index of the uncoated filet sample increased significantly compared to the initial value by 8 times. In chicken filet samples in alginate films coated with a wrapping method, the redness increased by 4.5 times; when coated by immersion, by 6.4 times, regardless of the introduction of protein hydrolysate. For the sample wrapped in a polyethylene film, the indicators of lightness (L) and redness (a*) did not change significantly. Oxidative reactions of lipids lead to the formation of metmyoglobin (yellow or brown), which affects a decrease in the value of lightness (L) and an increase in redness (a*) during cold storage. The substances with antioxidant properties included in the bioactive coating reduce oxidation reactions preventing a rapid decrease in the lightness index values (L).

The values of the yellowness index (b*) of all chicken filet samples gradually increased during storage, which may be due to the enzymatic browning reaction of phenolic components. Increased yellowness is also associated with the accumulation of myoglobin oxidation products, as well as with a high level of lipid oxidation [43]. Poultry meat tends to turn yellow during storage as the quality decreases. The results of the studies showed a noticeable increase in the yellowness index (b*) by 7 days of storage compared with the initial values for all filet samples. However, the uncoated control sample had higher yellowness values (b*) (*p* < 0.05). Experimental data confirmed that wrapping chicken meat in an alginate coating made it possible to effectively reduce the negative change in the color characteristics of poultry meat by preventing oxidative processes. The changes in the appearance of chicken meat samples in different types of packaging during storage are presented in Figure 5.

### 3.6. Rheological Characteristics

The type of packaging used for poultry meat can significantly influence its rheological characteristics, including elasticity, plasticity and deformation properties. In various studies, the authors have investigated how different packaging materials and packaging technologies affect deformation behavior of meat. Thus, studies have shown that modified atmosphere packaging effectively can preserve the structural integrity and elasticity of poultry meat compared to traditional packaging methods [44]. Vacuum packaging reduces moisture loss from muscle tissue, minimizes oxidative processes and inhibits microbial growth, while maintaining the plasticity of poultry meat [45].

In our research, it was found that on the third day of storage in poultry meat, both plastic and elastic deformations were significantly reduced when using packaging materials based on bioactive alginates due to the high moisture and vapor permeability of these films. At the same time, the elastic-plastic features of meat samples both in polyethylene film and without packaging have not changed significantly compared to the initial values.

On the seventh day of storage, more significant differences in the rheological characteristics of poultry meat stored in different packaging materials were observed. Thus, samples without packaging, due to intensive evaporation of moisture, became stiffer, with the lowest reductions in both plastic and elastic properties of muscle tissue. The opposite tendency was registered for meat in PET film. Here, due to low moisture permeability of the packaging material, we observed surface moistening, muscle softening and loss of elastic properties, respectively, the highest plastic and lowest elastic deformation for WF-PF sample were noticed (Table 5; Appendix A).

The ability of alginate films with protein hydrolysates to preserve the elastic and plastic properties of muscle tissue has been proven, due to the effectiveness of bioactive components in decelerating oxidative damage and microbiological deterioration, as well as in protecting the protein structure from degradation by oxidative processes and enzymatic activity [15,46].

Thus, elastic deformation for chicken meat samples WF-AF + PH and IF-AF + PH was significantly higher by 51.2–71.8%, and plastic deformation was significantly lower by 31.3–31.5% compared to samples in polyethylene film.

In summary, it was found that the composition of the packaging materials had a significant effect on the preservation of meat structure, and the use of bioactive compounds effectively prevented undesirable changes in the elastic and plastic deformations of poultry meat. In addition, it was demonstrated that the method of packaging also influenced the stability of rheological properties, when packing by wrapping, elastic-plastic properties were preserved more effectively than when packing by immersion. The results obtained in our study have been confirmed by the data of other researchers [47,48]. It has been proved that active packaging materials, such as bionanocomposites, from chitosan and sodium montmorillonite, in combination with ginger essential oil demonstrated the ability to reduce microbial growth and inhibit lipid oxidation, thereby improving the elasticity of meat and its structural integrity [49].

## 4. Conclusions

Bioactive packaging is essential for the transportation and storage of poultry meat as it helps maintain food quality, extends shelf life, and protects against unfavorable environmental factors and nutrients loss. The purpose of this study was to evaluate the potential of alginate-based packaging materials with the incorporation of protein hydrolysates to improve the safety and quality of chicken meat during storage. Physicochemical parameters, including weight loss, oxidation, and microbiological indicators, as well as color characteristics of chicken meat that was packaged in bioactive films, were determined.

The moisture content values were different when packing chicken meat samples in various types of films. A significant increase in moisture content for samples in polyethylene films (by 10.5%), and a decrease for samples that were wrapped in alginate-based films (by 3.1–5.3%) were observed. Meat samples in alginate films applied by immersion method demonstrated insignificant fluctuations in moisture content. The highest mass losses were found for the sample without packaging material, up to 20.4% and for samples wrapped in alginate films (15.9–17.9%). When packing meat samples by immersion method, a stable gradual decrease in weight (8.5–9.1%) was observed. On the 7th day of storage, the pH value of the control sample reached 6.55, while the samples for which alginate-based packaging materials with protein hydrolysate were applied showed more stable results (pH level 6.0–6.15).

The most pronounced oxidative processes were observed in the control meat sample. The application of alginate-based films with bioactive protein hydrolysate led to a significant reduction in fatty peroxide value by 52.9–56.2%, compared to the control. These films also showed an inhibitory effect on bacterial growth with a MAFAnM content in the meat samples packed in films with bioactive hydrolysate being 3.5–5 times lower than in the control sample. The highest microbial contamination was observed in a sample packed in polyethylene film. All types of packaging materials inhibited hydrolytic processes leading to a reduction in fatty acid content, and the lowest fatty acid values were detected for samples packaged in alginate-based films with protein hydrolysate. Substances with antioxidant properties presented in bioactive films can reduce oxidation reactions and prevent a rapid decrease in the lightness index values. According to the experimental data, chicken meat wrapped in alginate-based films maintains more stable color characteristics, due to the preventive oxidative processes. It was found that the composition of the packaging materials had a significant effect on the preservation of meat structure, and the use of bioactive compounds effectively prevented undesirable changes in the elastic and plastic properties of poultry meat.

As a result of the study, experimental data confirmed that chicken meat packed in alginate-based film with the inclusion of protein hydrolysates contributes to the effective prevention of negative alterations in the characteristics of poultry meat by inhibiting undesirable oxidative and microbiological processes. Research in the field of application of alginate-based films with the inclusion of bioactive components is quite beneficial, as it allows to develop advanced cost-effective approaches to maintain the quality and safety of chicken meat and extend its shelf life.

## Figures and Tables

**Figure 1 polymers-16-03430-f001:**
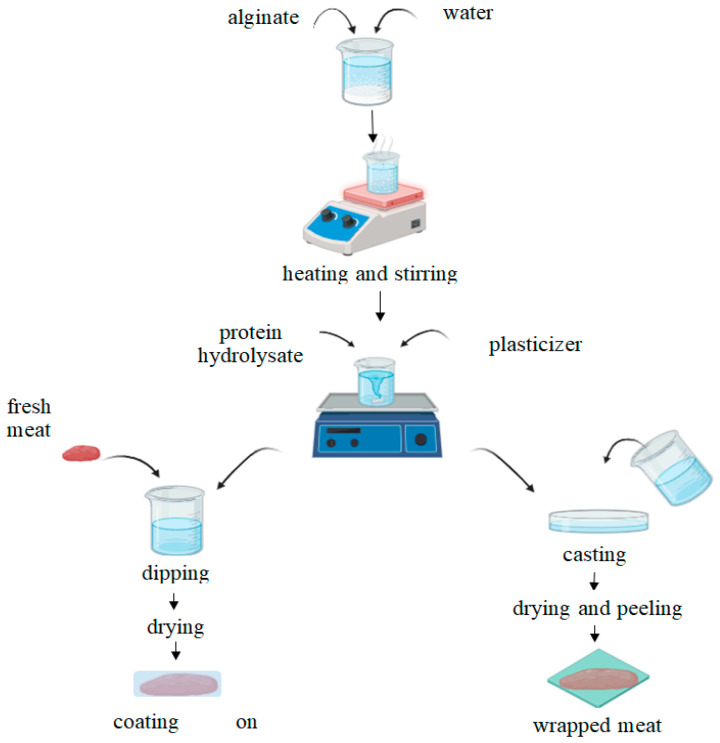
Scheme for the production of bioactive alginate-based packaging materials.

**Figure 2 polymers-16-03430-f002:**
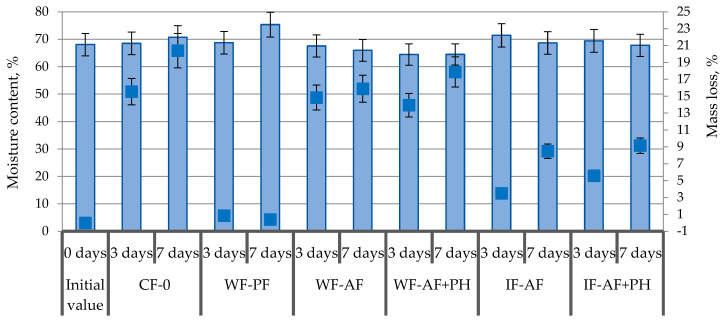
Dynamics of moisture content (light blue) and weight loss (dark blue) in poultry meat samples during storage.

**Figure 3 polymers-16-03430-f003:**
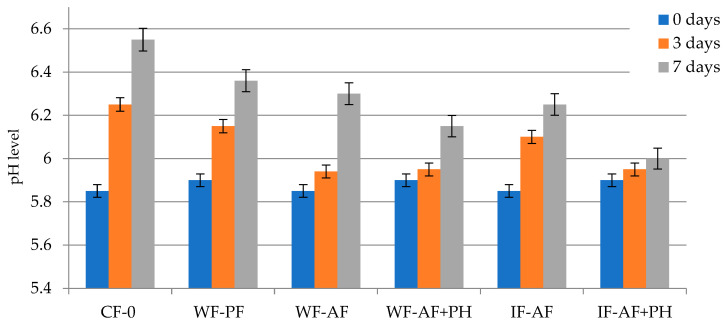
Dynamics of the pH level in chicken meat during refrigerated storage.

**Figure 4 polymers-16-03430-f004:**
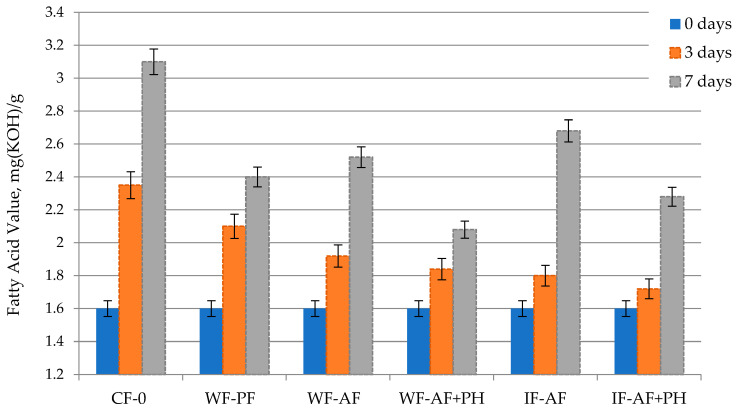
Dynamics of the Fatty Acid Value in chicken meat during refrigerated storage.

**Figure 5 polymers-16-03430-f005:**
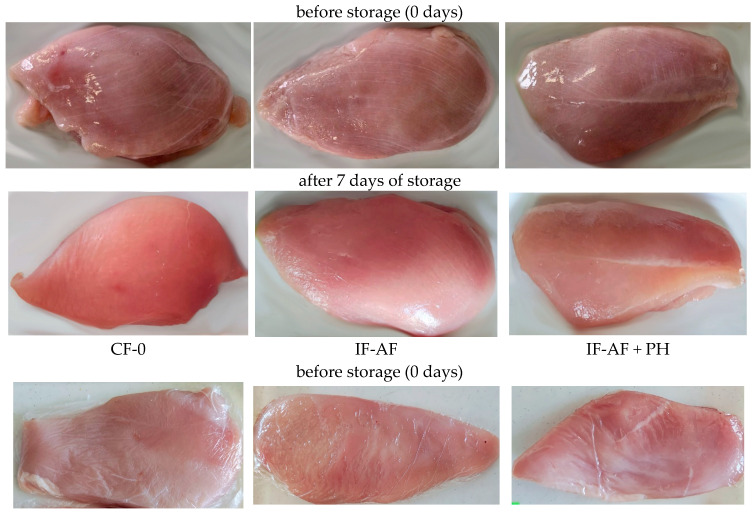
Changing the appearance of chicken meat samples during storage at temperature (4 ± 2) °C.

**Table 1 polymers-16-03430-t001:** Code and characteristics of chicken meat samples.

Indicators	Code of Chicken Meat Sample/Sample Characteristic					
	CF-0	WF-PF	WF-AF	WF-AF + PH	IF-AF	IF-AF + PH
Type of meat	chicken filetfilet	chicken filetfilet	chicken filetfilet	chicken filetfilet	chicken filetfilet	chicken filetfilet
Type of packaging materials	without packaging	PET film	alginate-based films	alginate-based films	alginate-based films	alginate-based films
Packing technique	not applicable	wrapping	wrapping	wrapping	immersion	immersion
Bioactive compound	not applicable	not applicable	not applicable	protein hydrolysate	not applicable	protein hydrolysate

**Table 2 polymers-16-03430-t002:** Lipid oxidation indices in chicken filet during storage.

Time (Days)	CF-0	WF-PF	WF-AF	WF-AF + PH	IF-AF	IF-AF + PH
Fatty Peroxide Value, mmol (12O_2_)/kg						
0	1.2 ± 0.02 ^a^	1.3 ± 0.02 ^a^	1.3 ± 0.02 ^a^	1.2 ± 0.02 ^a^	1.2 ± 0.02 ^a^	1.3 ± 0.02 ^a^
3	2.8 ± 0.04 ^a^	2.5 ± 0.03 ^a^	1.9 ± 0.03 ^b^	1.8 ± 0.03 ^b^	2.5 ± 0.03 ^a^	1.9 ± 0.03 ^b^
7	5.1 ± 0.05 ^a^	2.5 ± 0.03 ^c^	2.8 ± 0.04 ^bc^	2.2 ± 0.02 ^d^	3.0 ± 0.04 ^b^	2.4 ± 0.03 ^c^
Fatty Iodine Value, g/100 g						
0	21 ± 0.20 ^a^	21 ± 0.20 ^a^	20 ± 0.25 ^a^	20 ± 0.25 ^a^	21 ± 0.20 ^a^	21 ± 0.20 ^a^
3	15 ± 0.24 ^b^	18 ± 0.25 ^ab^	16 ± 0.15 ^b^	19 ± 0.30 ^a^	18 ± 0.25 ^ab^	19 ± 0.30 ^a^
7	8 ± 0.12 ^c^	15 ± 0.25 ^ab^	12 ± 0.12 ^b^	17 ± 0.24 ^a^	15 ± 0.25 ^ab^	18 ± 0.24 ^a^

The data are the averaged values of triplicate repetition (±standard deviation); a, b, c, d—values with different letters in the same row differ at *p* ≤ 0.05.

**Table 3 polymers-16-03430-t003:** Microbiological Indicators of chicken filet during storage.

Storage Period (Days)	CF-0	WF-PF	WF-AF	WF-AF + PH	IF-AF	IF-AF + PH
Mesophilic aerobic and facultative anaerobic microorganisms, Log CFU/g meat						
0	3.40 ^a^	3.38 ^a^	3.40 ^a^	3.40 ^a^	3.38 ^a^	3.40 ^a^
3	4.53 ^b^	5.26 ^a^	4.57 ^b^	4.40 ^b^	4.70 ^b^	4.36 ^b^
7	5.32 ^b^	5.77 ^a^	4.98 ^c^	4.78 ^cd^	5.00 ^c^	4.60 ^d^
Aerobic psychotropic microorganisms, Log CFU/g meat						
0	2.18 ^a^	2.14 ^a^	2.17 ^a^	2.18 ^a^	2.16 ^a^	2.15 ^a^
3	3.91 ^b^	4.11 ^a^	3.85 ^b^	3.79 ^b^	3.82 ^b^	3.60 ^b^
7	5.07 ^b^	4.80 ^a^	4.41 ^c^	4.36 ^cd^	4.20 ^c^	4.14 ^d^
*Enterobacteriaceae*, Log CFU/g meat						
0	2.01 ^a^	1.97 ^a^	2.02 ^a^	1.98 ^a^	2.01 ^a^	2.02 ^a^
3	3.84 ^b^	3.81 ^a^	3.70 ^b^	3.63 ^b^	3.57 ^b^	3.59 ^b^
7	4.72 ^b^	4.94 ^a^	4.09 ^c^	4.01 ^c^	3.96 ^c^	3.91 ^d^

The data are the averaged values of triplicate repetition (±standard deviation); a, b, c, d—values with different letters in the same row differ at *p* ≤ 0.05.

**Table 4 polymers-16-03430-t004:** Color characteristics of chicken meat during refrigerated storage.

Indicators	CF-0	WF-PF	WF-AF	WF-AF + PH	IF-AF	IF-AF + PH
before storage (0 days)						
L*	48.10 ± 1.00 ^a^	48.14 ± 1.10 ^a^	48.12 ± 1.10 ^a^	48.14 ± 1.00 ^a^	48.15 ± 1.10 ^a^	48.14 ± 1.10 ^a^
a	0.77 ± 0.05 ^c^	0.75 ± 0.05 ^a^	0.76 ± 0.06 ^b^	0.75 ± 0.05 ^b^	0.76 ± 0.06 ^c^	0.78 ± 0.06 ^c^
b	1.75 ± 0.50 ^b^	1.73 ± 0.45 ^b^	1.74 ± 0.45 ^b^	1.73 ± 0.40 ^b^	1.75 ± 0.50 ^c^	1.73 ± 0.45 ^c^
ΔE	45.55 ± 0.85 ^b^	45.53 ± 0.90 ^a^	45.50 ± 0.80 ^b^	45.53 ± 0.90 ^b^	45.50 ± 0.85 ^a^	45.53 ± 0.90 ^a^
Croma	2.6 ± 0.15 ^c^	2.5 ± 0.10 ^a^	2.4 ± 0.12 ^b^	2.5 ± 0.10 ^c^	2.5 ± 0.15 ^c^	2.4 ± 0.10 ^c^
after 3 days of storage						
*L	37.13 ± 0.80 ^b^	51.33 ± 1.10 ^a^	40.11 ± 0.85 ^ab^	39.84 ± 0.80 ^ab^	48.43 ± 0.90 ^a^	44.63 ± 0.80 ^a^
**a	3.86 ± 0.45 ^b^	1.91 ± 0.24 ^a^	3.64 ± 0.36 ^a^	2.88 ± 0.30 ^a^	2.64 ± 0.25 ^b^	2.85 ± 0.22 ^b^
***b	5.88 ± 0.56 ^a^	1.78 ± 0.15 ^b^	2.46 ± 0.25 ^a^	1.41 ± 0.10 ^b^	3.44 ± 0.35 ^b^	3.4 ± 0.30 ^b^
ΔE	49.3 ± 1.20 ^ab^	41.15 ± 1.30 ^a^	46.96 ± 1.44 ^b^	52.04 ± 1.45 ^a^	41.66 ± 1.10 ^a^	45.29 ± 1.30 ^a^
Croma	9.74 ± 0.65 ^b^	3.69 ± 0.35 ^a^	5.1 ± 0.40 ^a^	4.29 ± 0.35 ^b^	6.08 ± 0.54 ^b^	6.25 ± 0.50 ^b^
after 7 days of storage						
L	31.5 ± 0.80 ^c^	49.81 ± 1.05 ^a^	37.49 ± 0.85 ^b^	37.49 ± 0.80 ^b^	36.78 ± 0.80 ^b^	34.75 ± 0.76 ^b^
a	6.12 ± 0.55 ^a^	0.93 ± 0.07 ^a^	3.46 ± 0.25 ^a^	3.46 ± 0.30 ^a^	4.93 ± 0.38 ^a^	5.17 ± 0.65 ^a^
b	5.54 ± 0.60 ^a^	2.6 ± 0.32 ^a^	2.61 ± 0.26 ^a^	2.61 ± 0.25 ^a^	4.42 ± 0.55 ^a^	4.49 ± 0.50 ^a^
ΔE	53.01 ± 1.80 ^a^	42.83 ± 1.25 ^a^	52.61 ± 1.65 ^b^	52.61 ± 1.65 ^a^	50.04 ± 1.50 ^a^	51.76 ± 1.45 ^a^
Croma	11.66 ± 0.80 ^a^	3.53 ± 0.24 ^a^	6.07 ± 0.45 ^a^	6.07 ± 0.40 ^a^	9.35 ± 0.60 ^a^	9.66 ± 0.65 ^a^

*L—lightness, **a—redness, ***b—yellowness, ΔE—Total Color difference), and Chroma—Color Intensity. The data are the averaged values of triplicate repetition (±standard deviation); a, b, c—values with different letters in the same row differ at *p* ≤ 0.05.

**Table 5 polymers-16-03430-t005:** Rheological characteristics of chicken meat during refrigerated storage.

Sample Code	Rheological Indicators		
	General Deformation, mm	Plastic Deformation, mm	Elastic Deformation, mm
Before storage (0 days)			
–	12.11 ± 0.493 ^a^	7.19 ± 0.549 ^b^	4.92 ± 0.122 ^a^
After 3 days of storage			
CF-0	10.98 ± 0.449 ^b^	7.15 ± 0.547 ^b^	3.83 ± 0.311 ^b^
WF-PF	10.78 ± 0.529 ^b^	6.74 ± 0.598 ^b^	3.97 ± 0.119 ^b^
WF-AF	9.88 ± 0.496 ^b^	5.76 ± 0.230 ^c^	4.13 ± 0.117 ^ab^
WF-AF + PH	8.10 ± 0.420 ^c^	4.83 ± 0.286 ^d^	3.28 ± 0.134 ^b^
IF-AF	8.73 ± 0.492 ^c^	4.94 ± 0.236 ^d^	3.79 ± 0.206 ^b^
IF-AF + PH	9.55 ± 0.411 ^b^	6.06 ± 0.268 ^bc^	3.50 ± 0.223 ^b^
After 7 days of storage			
CF-0	6.82 ± 0.517 ^d^	4.86 ± 0.345 ^d^	1.96 ± 0.128 ^d^
WF-PF	11.84 ± 0.311 ^a^	9.24 ± 0.320 ^a^	1.60 ± 0.04 ^d^
WF-AF	9.58 ± 0.445 ^b^	6.35 ± 0.418 ^b^	2.60 ± 0.118 ^c^
WF-AF + PH	8.81 ± 0.403 ^c^	6.33 ± 0.383 ^b^	2.75 ± 0.08 ^c^
IF-AF	8.33 ± 0.316 ^c^	6.20 ± 0.360 ^b^	2.33 ± 0.06 ^cd^
IF-AF + PH	8.58 ± 0.205 ^c^	6.35 ± 0.210 ^b^	2.42 ± 0.07 ^c^

The data are the averaged values of triplicate repetition (±standard deviation); a, b, c, d—values with different letters in the same column differ at *p* ≤ 0.05.

## Data Availability

Data will be made available on request to the corresponding author.

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
