# Peer review of "Effect of Bioactive Packaging Materials Based on Sodium Alginate and Protein Hydrolysates on the Quality and Safety of Refrigerated Chicken Meat"

_polymers, 2024, doi:10.3390/polym16233430_

Round 1
Reviewer 1 Report
Comments and Suggestions for Authors
This study evaluated the effectiveness of innovative alginate-based packaging materials enriched with protein hydrolysates in enhancing the safety and quality of chicken meat during storage. The results are presented but need more clarity and coherence. Avoid listing data values without context or interpretation. Instead, focus on the trends and their implications. In conclusion, add/Highlight the overall benefits of alginate-based films in preserving chicken meat quality.
With the following points with major revision, the paper has been recommended for next level.
1. Rewrite the abstract and add more data
2. Line No. 34: Give the latest citation and rewrite the line
3. Why authors choose chicken meat (fillet) as a research material?
4. Protein hydrolysate was obtained by microbial fermentation of broiler chicken gizzards with propionic acid bacteria: explain these lines??
5. Line 142: why heated at 42oC?
6. 2.4.4: Add more quality parameters which will help the readers to understand this work nicely.
7. Section 2.4.9: The data were analyzed via one-way ANOVA analysis: Why?? Why not with some advanced tools for clear results?
8. Why the Fatty Acid Value in chicken meat decreases during refrigerated storage
Author Response
Dear Sir/Madam,
We appreciate the reviewer for putting a lot of time into reading, editing, and commenting on our manuscript. We are very grateful for the critical comments and constructive suggestions that helped us improve the quality of the manuscript.
We have revised our manuscript based on the reviewer's comments. All corrections in the manuscript are highlighted in yellow.
Specific response to the reviewers’ comments you could find in the attachment.

Reviewer 2 Report
Comments and Suggestions for Authors
Polymers Manuscript 3272947
The article "Effect of Bioactive Packaging Materials Based on Sodium Alginate and Protein Hydrolysates on the Quality and Safety of Refrigerated Chicken Meat" by Svetlana Merenkova and Oksana Zinina describes a study of alginate-based packaging materials with the incorporation of protein hydrolysates to improve the safety and quality of chicken meat during storage. Physicochemical parameters were determined, including weight loss, oxidation, micro-biological indicators, and the color characteristics of chicken meat packaged in bioactive films. The paper is sufficiently interesting to warrant publication in Polymers. The manuscript is clear and written in good English. However, some corrections are necessary. Comments:
1. Line 134: better sentence: Protein hydrolysate was obtained by microbial fermentation of broiler chicken gizzards with propionic acid bacteria. The technology is described in [26, 27].
2. Line 150: What about the thickness of the films?
3. Line 153: better sentence: The chicken meat samples (for codes, see Table 1) were ......
4. Line 155; remove in this sentence: (Table 1)
5. Line 158: This must be: Table 1
6. Line 200: Use increasing order of numbers for the equations: (1), (2), (3) etc
7. Line 204: 0.1mol/dm3 must be 0.1 mol/dm3
8. Line 214: Again, use increasing order of numbers for the equations: (1), (2), (3) etc
9. Line 223: explain: Viisa reagent
10. Line 224: (0.1 mol / dm3) must be (0.1 mol/dm3)
11. Line 230: 100 g / 230 dm3 must be 100 g/230 dm3
12. Line 234: .... FI), g / 100 g is .... must be .... FI), g/100 g, is ....
13. Line 250: renumber the equations, see comments 5 and 7
14. Line 306: define light blue and dark blue in the legend of Figure 2
15. Line 315: pH values are not in (Fig.2) but in (Fig. 3)
16. Line 345: In Table 2: g / 100 g must be g/100 g
17. Line 375: ........ causing their death. Literature references are needed.
18. Line 384 and 395: exchange the ref numbers, also in the ref list.
18. Line 410: define L, a, b, ΔE, and Croma in the Table legend or refer to Section 2.4.7. Also, better to have Table 4 in total on one page.
20. Line 441: Is the upper part of Figure 5 correct? Put storage temperature in the figure legend.
21. Line 483: ......... of other researchers [references needed].
Author Response
Dear Sir/Madam,
We appreciate the reviewer for putting a lot of time into reading, editing, and commenting on our manuscript. We are grateful for the critical comments and constructive suggestions that helped us improve the quality of the manuscript.
We have revised our manuscript based on the reviewer's comments. All corrections in the manuscript are highlighted in yellow.
Specific responses to the reviewers’ comments could be find in the attachment.

Round 2
Reviewer 1 Report
Comments and Suggestions for Authors
All the suggestions are properly incorporated by the Author. In this version the author has given more importance to technical accuracy and hence the manuscript may be accepted.